# Feasibility of the 30 s Sit-to-Stand Test in the Telehealth Setting and Its Relationship to Persistent Symptoms in Non-Hospitalized Patients with Long COVID

**DOI:** 10.3390/diagnostics13010024

**Published:** 2022-12-21

**Authors:** Rodrigo Núñez-Cortés, Cristina Flor-Rufino, Francisco Miguel Martínez-Arnau, Anna Arnal-Gómez, Claudia Espinoza-Bravo, David Hernández-Guillén, Sara Cortés-Amador

**Affiliations:** 1Physiotherapy in Motion Multispeciality Research Group (PTinMOTION), Department of Physiotherapy, University of Valencia, 46010 Valencia, Spain; 2Department of Physical Therapy, Faculty of Medicine, University of Chile, Santiago 8240000, Chile; 3Department of Physiotherapy, Faculty of Physiotherapy, University of Valencia, 46010 Valencia, Spain; 4Day Hospital Unit, Hospital Clínico la Florida, Santiago 8240000, Chile; 5Group of Physiotherapy in the Aging Process: Social and Health Care Strategies (PT_AGE), Department of Physiotherapy, Faculty of Physiotherapy, University of Valencia, 46010 Valencia, Spain

**Keywords:** telemedicine, functional capacity, rehabilitation, SARS-CoV-2, pain

## Abstract

Fatigue, dyspnea and pain are the main limitations of patients with long COVID. The aim of this study was to determine the feasibility of the 30 s sit-to-stand (30s-STS) test in the telehealth setting and its relationship to persistent symptoms in a sample of non-hospitalized patients with long COVID. A cross-sectional study was conducted in community patients with long COVID. Data collection and assessments were performed by videoconference and consisted of the fatigue assessment scale (FAS), London activity of daily living scale (LCADL), post-COVID-19 functional status (PCFS) and European quality of life questionnaire (EQ-5D-5L), including the pain/discomfort dimension. The 30s-STS test was performed using a standardized protocol adapted for remote use, and the modified Borg scale (0–10) was used to assess dyspnea and lower limb fatigue immediately after the test. The feasibility of the 30s-STS test was assessed by the proportion of eligible participants who were able to complete the test. Safety was assessed by the number of adverse events that occurred during the test. Seventy-nine participants were included (median age: 44 years, 86.1% women). Performance in the 30s-STS test was 11.5 ± 3.2 repetitions with 60.8% of the sample below reference values. All eligible participants were able to complete the test. No adverse events were reported during the evaluation. Participants with lower 30s-STS performance had more fatigue and dyspnea, worse quality of life, more severe pain/discomfort, and worse functional status (*p* < 0.05). A significant correlation was obtained between LCADL and dyspnea, reported on the Borg scale (0–10) post 30s-STS (r = 0.71; *p* < 0.001). In conclusion, the 30s-STS test proved to be a feasible test to implement in the telehealth setting and is related to fatigue, dyspnea, quality of life and pain in non-hospitalized patients with long COVID. Clinicians may use this test when assessment of the physical sequelae of COVID-19 in the face-to-face setting is not possible.

## 1. Introduction

Since the onset of the COVID-19 pandemic in December 2019, more than 640 million cases have been confirmed worldwide according to the World Health Organization (WHO) [1]. In addition, the restrictions adopted to control the quick spread of COVID-19 have had negative consequences on the overall health of the population (e.g., the practice of physical activity levels have been affected) [2]. Previous research has reported that 81% of COVID-19 cases show a mild presentation of the disease, 14% moderate and the remaining 5% trigger a critical situation [3,4]. Post-COVID-19 sequelae may be present in more than 60% of infected people [5], with female sex being a risk factor for the development of some persistent symptoms [6].

In recent months, research has focused on the post-infection stages, as it has been reported that, in some patients, symptomatology may reappear and persist for months or even years after infection [7,8]. This persistent condition is named “Long COVID” [6] and it can affect different organs and body systems, with a wide range of signs and symptoms. The most commonly reported symptoms are fatigue, dyspnea and pain [9], with no differences between hospitalized and non-hospitalized patients [5]. These sequelae can affect physical performance, activities of daily living, and lead to a loss of health-related quality of life [10]. Thus, an evaluation and follow-up of individuals who have suffered COVID-19 is considered desirable in order to detect sequelae and implement treatment if necessary [11,12].

In this context, the assessment of patients’ functional performance after COVID-19 has become a challenge for clinicians’ decision-making. The 6 min walk test (6MWT) is considered the gold standard for functional performance assessment. However, the 6MWT requires technical performance conditions that are not easy to meet in the telerehabilitation setting, such as a 20–30 m corridor [13]. In contrast, the 30 s sit-to-stand (30s-STS) is a quick and easy-to-use, low-cost clinical test of functional capacity, which has been validated in vulnerable populations such as older adults [14] or oncological patients [15]. The 30s-STS is a time-based assessment in which participants are asked to stand and sit from a chair as many times as possible for 30 s with their arms crossed over their chest [14]. In general, the 30s-STS is better tolerated than the 1-min STS [16], and performance requires greater cardiorespiratory endurance than the five times STS [17].

Due to the outbreak of COVID-19, many rehabilitation programs were adapted from face-to-face to remote models [18,19]. Thus, compared to face-to-face programs, telehealth programs can eliminate geographic and socioeconomic barriers by improving access for participants in rural and transportation-challenged areas, and are a suitable alternative for clinical assessment and intervention during pandemics [18,20,21]. For example, a recent meta-analysis identified that home-based cardiac rehabilitation significantly improves functional capacity and health-related quality of life, compared to usual care, being a potential alternative for patients who are not suitable for in-center cardiac rehabilitation [22]. In addition, a recent systematic review concluded that the risk of adverse events during home rehabilitation appears to be very low in cardiac patients [23]. In this context, the STS test may also be performed safely at home, provided that patients are not at risk of desaturation [24]. Traditionally, the 30s-STS has been used to assess lower limb strength, muscle power or physical function [25,26,27]. However, since the performance of the test requires some cardiorespiratory demand [17], it could also be an alternative to assess lower limb fatigue and dyspnea on physical exertion, which are both very prevalent symptoms in patients with long COVID. Therefore, the use of the 30s-STS in a telehealth context could be interesting, especially for assessing patients who did not have early access to rehabilitation programs, such as people who have suffered a mild COVID-19 infection and yet experienced persistence of COVID-19 symptoms months after the initial episode. Consequently, the objective of this study was to determine the feasibility of the 30s-STS test in the telehealth setting and its relationship to persistent symptoms such as fatigue, dyspnea and pain in a sample of non-hospitalized patients with long COVID.

## 2. Materials and Methods

### 2.1. Design and Participants

We conducted a cross-sectional study that collected data from consecutive community patients with long COVID admitted to a telerehabilitation program implemented at the University of Valencia (Valencia, Spain) between October 2021 and May 2022. Participants were recruited through social networks and by contact with long COVID associations in the autonomous communities of Comunidad Valenciana, Madrid, Castilla la Mancha, Cataluña, Galicia, Cantabria and Aragon. Inclusion criteria were as follows: (i) age between 20 and 60 years old; (ii) positive PCR test results from nasal and pharyngeal swab sample; (iii) presence of at least one of the following persistent COVID-19-related symptoms: fatigue, dyspnea, or functional limitation for at least 6 weeks after infection; (iv) having a device with Internet access (e.g., smartphone, computer or tablet). Exclusion criteria were: (i) severe case of COVID-19 (i.e., history of hospitalization, severe pneumonia or pulmonary thromboembolism); (ii) other concomitant acute or chronic pulmonary or cardiac pathologies; (iii) presence of more severe symptoms requiring monitorization by clinical staff (i.e., desaturation on exertion, unsteadiness, hemodynamic instability).

This study was approved by the Ethics Committee of the University of Valencia (Registration number: 15737788), and all patients provided written informed consent. The research was conducted in accordance with the ethical principles of the Declaration of Helsinki. This study was conducted in accordance with the Guidelines for Strengthening the Reporting of Observational Studies in Epidemiology (STROBE) [28].

### 2.2. Data Collection

The evaluation was conducted by 1:1 videoconference in real time using Zoom Communication software (Zoom Video Communications, Inc., San Jose, CA, USA). Data were collected by four PhD physiotherapists (C.F-R., F.M.M-A., A.A.-G. and D.H-G.) with more than five years of clinical experience. All evaluators received prior training to standardize the evaluation. The assessment was performed in the patient’s environment, using the device of their choice (computer, smartphone or tablet), and data protection was ensured by storing the information anonymously, with access restricted to research staff only. Data regarding age, sex, time post-infection, symptoms related to long COVID, and smoking history were collected by structured interview.

The level of fatigue was assessed using the fatigue assessment scale (FAS) [29], which consists of 10 items evaluating both physical and mental fatigue, with 5 questions, respectively. Each item is scored on a scale of 1 “never” to 5 “always,” with a higher score (which ranged between 10 and 50) indicating a higher level of fatigue. This instrument has proven to be valid and reliable for fatigue assessment [29].

Dyspnea was assessed using the Spanish version of the London chest activity of daily living scale (LCADL) [30], which is a valid questionnaire that evaluates the degree of limitation in activities of daily living due to dyspnea in patients with chronic respiratory diseases. The LCADL comprises 15 items that consider self-care, household, physical and leisure activities, and each question is scored from 0 to 5. A higher score indicates a greater degree of limitation in activities of daily living due to dyspnea [30].

Health-related quality of life was assessed using the 5-dimensional European quality of life questionnaire (EQ-5D-5L) [31], which provides an index score ranging from 0 (death) to 1 (full health), and a self-reported rating of current general health status based on a visual analogue scale ranging from 0 “the worst health you can imagine” to 100 “the best health you can imagine”. Pain or discomfort were rated using a 5-choice categorical scale: (1) “No pain or discomfort”; (2) “Slight pain or discomfort”; (3) “Moderate pain or discomfort”; (4) “Severe pain or discomfort”; (5) “Extreme pain or discomfort”.

Finally, functional status was assessed with the Spanish version of the Post-COVID-19 functional status scale (PCFS) [32], which is a 6-grade ordinal scale: grade 0 (no functional limitations); grade 1 (negligible functional limitations); grade 2 (slight functional limitations); grade 3 (moderate functional limitations); grade 4 (severe functional limitations); and grade 5 (death). The feasibility of the 30s-STS test was assessed by the proportion of eligible participants who were able to complete the test. Reasons for not completing the test were also recorded. Safety was assessed by the number of adverse events of any type (serious or minor) that occurred during the performance of the 30s-STS test.

The 30s-STS test was performed in the participant’s home environment using a standardized protocol adapted for remote use [15]. First, the clinician explained the 30s-STS test and ensured that the patient understood how to perform it. Adequate Internet connection was also checked. Then, participants were instructed to place a sturdy chair against the wall. Participants were asked to position themselves in the center of the device’s camera view to obtain the best visibility for the clinician. If available, participants were asked to use a chair without armrests. The 30s-STS test was performed only once, since it is considered to have good test–retest reliability [33]. Patients were instructed to cross their arms over their chest and complete as many standing cycles as possible in 30 s. The instructions were to stand until fully upright and then sit until the buttocks touched the chair, without aid of their hands [27]. During the test they were verbally encouraged [27]. To compare functional performance in the 30s-STS test with reference values in healthy populations, the sex- and age-specific centile curves reported by Warden et al. [34] were used. To categorize low and normal functional performance in the 30s-STS test, the lower limit of the standard deviation of the mean number of repetitions according to age and sex was used as the cutoff point. The modified Borg scale (0–10) was used to assess dyspnea and lower limb fatigue immediately after the 30s-STS test [35]. The Borg scale score ranges from 0 to 10, where 0 corresponds to the absence of dyspnea or physical exertion and 10 corresponds to the maximum degree of dyspnea or physical exertion.

### 2.3. Statistical Analysis

Sample size calculation was performed with G*Power, version 3.1.9.2 (Universität Düsseldorf, Germany). A moderate effect size (d = 0.7) was estimated from a clinically relevant 4-point difference in fatigue (FAS score) [36] (difference between two independent means, n1 ≠ n2). Considering a statistical power of 80%, two tails, and α err prob = 0.05, the minimum required sample size was 68 patients. Statistical analysis was performed with Statistical Package for the Social Sciences (SPSS) version 22.0 (IBM Corporation, Armonk, NY, USA). Normality of the data was determined with the Shapiro–Wilk test. Considering the distribution of the data, parametric or nonparametric, the results were presented as mean and standard deviation or as median and interquartile range (IQR), respectively. Comparison between the low and normal performance groups in the 30s-STS according to baseline values was performed using the chi-square test for categorical variables (sex, PCFS and pain/discomfort), the Mann–Whitney U-test for nonparametric variables (post-infection time and dyspnea), and the independent samples *t*-test for variables with normal distribution. A correlation analysis using Spearman’s correlation coefficient was applied to assess the association between lower limb fatigue and dyspnea, measured with the Borg scale (0–10) post 30s-STS, and the FAS and LCADL, respectively. The significance level was set at 0.05 for all statistical analyses.

## 3. Results

A total of 79 participants met the eligibility criteria (Figure 1).

The median age was 44 (range: 24–52) years, and 68 (86.1%) participants were women. Time passed after COVID-19 infection ranged from 2 to 28 months, with a median of 17 months. In the total sample, 72.2% of cases had moderate to extreme pain, with high levels of dyspnea and fatigue (Table 1).

Performance in the 30s-STS test was 11.5 ± 3.2 repetitions and 48 (60.8%) cases performed below the reference values according to age and sex (Figure 2).

No adverse events were reported during the evaluation. All patients were able to complete the test; only two participants reported mild dizziness at the end of the test. Significant differences in LCADL (*p* = 0.001) and FAS (*p* = 0.004) were obtained when comparing the low and normal functional performance groups in the 30s-STS according to reference values. Moreover, participants with lower 30s-STS performance had worse quality of life on the EQ5D index score (mean difference = −0.22, 95% confidence interval: −0.32 to −0.14, *p* < 0.001), on the visual analogue scale (mean difference = −13.9, 95% confidence interval: −21.1 to −6.7, *p* < 0.001), more severe pain/discomfort (χ^2^ = 13.1, *p* = 0.011), and more severe PCFS (χ^2^ = 11.1, *p* = 0.011). There were no differences with respect to age and time post infection (*p* > 0.05) (Table 2).

The median (IQR) fatigue and dyspnea reported on the Borg scale (0–10) after the 30s-STS test were 6.0 (5.0) and 3.0 (5.0), respectively. A low and significant correlation was obtained between FAS and lower limb fatigue reported on the Borg scale (0–10) post-30s-STS (r = 0.24, *p* = 0.034). A high and significant correlation was obtained between LCADL and dyspnea reported on the Borg scale (0–10) post-30s-STS (r = 0.71, *p* < 0.001). The distribution of dyspnea and fatigue data for each scale are shown in Appendix A.

## 4. Discussion

The 30s-STS test proved to be a feasible test to implement in the telehealth setting when assessing physical function and its relationship to persistent symptoms such as fatigue, dyspnea, quality of life and pain/discomfort in a sample of community patients with long COVID. Functional tests (e.g., 30s-STS) performed via teleassessment are reliable, valid and feasible for measuring the performance of healthy young adults in clinical practice [37]. Considering that all the cases included in our study were able to perform the test and no adverse events were recorded, our results indicate that the 30s-STS may also be an excellent option for telehealth assessment of the main symptoms of prolonged COVID (e.g., fatigue, dyspnea, functional impairment and pain), especially in pandemics, when equipment, time and space requirements may be limited. Therefore, rehabilitation clinicians may perform the 30s-STS test with confidence when they aim to identify cases with greater physical sequelae [15,24].

Our results are similar to those published by Bowman et al. [15], who evaluated the feasibility and safety of 30s-STS via telehealth in the oncology population, with a 94% test completion rate and no reported safety incidents. Furthermore, in this investigation they found a moderate correlation between the 30s-STS and self-reported physical activity level, providing evidence of convergent validity [15]. Interestingly, performance in the number of repetitions of the 30s-STS in our population (median = 11.5 repetitions) was lower than the one reported by Bowman et al. [15] (median 13.5 repetitions) even when our sample was, according to median age, 18 years younger. These findings show that the physical sequelae following COVID-19 were also significant in non-hospitalized cases, regardless of severity.

More than half of the cases had poor functional performance (i.e., 30s-STS below the reference values), which was associated with increased levels of fatigue, dyspnea and pain/discomfort. In fact, the difference between groups was greater than the minimal important difference established for FAS (4-point) [36], LCADLtotal (range: −2.1 and −5.9 points), and for LCADL%total (−2 and −6 points) in patients with chronic respiratory diseases (Table 2) [38]. On the other hand, the modified Borg scale used to assess lower limb fatigue and dyspnea after the 30s-STS test showed a significant association with the validated scales for these symptoms, FAS and LCADL, respectively. This test has been commonly used as an indicator of lower limb muscle strength/power in patients with chronic conditions [25,26,27]. However, taking into account that the number of repetitions in the 30s-STS has a moderate correlation with the distance walked in the six-minute walk test and therefore requires some physical and cardiorespiratory demand [17], our results indicate that this test may also be applied to assess symptoms of fatigue and dyspnea after physical exertion, which are very frequent in patients with long COVID [9].

Patients with low 30s-STS performance according to reference values also had worse quality of life and more severe functional status on PCFS. In addition, participants with low 30s-STS performance had a higher severity of pain/discomfort. In fact, more than 85% of them had moderate to extreme pain/discomfort, in contrast to participants with normal 30s-STS performance, of which only 51.6% had this condition. Persistent pain is one of the most common symptoms in long COVID and addressing it could be key to improving functional performance as well [39].

The COVID-19 pandemic has forced healthcare teams to innovate and implement new strategies for monitoring patients in need of rehabilitation [19]. Thus, the 30s-STS test can also be used to prescribe exercise in telehealth programs or when geographic or economic barriers prevent assessment of COVID-19 physical sequelae. For example, functional performance assessment through telehealth could improve access to follow-up of physical sequelae for people living in rural areas, having financial or transport problems, insufficient social assistance or being unable to take time off work, which are common barriers to rehabilitation programs [18]. Telehealth also presents an opportunity to safely benefit vulnerable populations (e.g., home-based rehabilitation). However, new approaches are needed to achieve a sense of connection similar to that of face-to-face care in terms of practical teaching, training and human connection [19]. Particularly, improving functional performance using the 30s-STS test has also been a goal of telerehabilitation programs for people with COVID-19 and post-COVID-19 conditions. For example, exercises performed through telerehabilitation can specifically improve performance in the 30s-STS test [40]. Thus, this test can be used for both screening and assessment of post-treatment changes. Additionally, a functional assessment test prior to the start of the rehabilitation program could allow an accurate exercise prescription at home [41]. The 30s-STS has the potential to be delivered safely through telehealth [15], reducing the delivery cost and improving patient access and autonomy [40,41,42].

The main limitation of this study was that the chair used for the assessment was not standardized for all participants, as it varied from household to household. This could limit comparison with baseline data and therefore the results should be interpreted with caution. In addition, the device for evaluation varied among each participant. On the other hand, we were unable to control cardiorespiratory variables such as oxygen saturation or heart rate. Finally, we recognize that there may be a selection bias because recruitment of participants was by disclosure and voluntary participation. However, our study has a pragmatic focus and may provide guidance to clinicians when assessing and prescribing exercise to their patients via telehealth. Future studies should evaluate the possibility of a more comprehensive assessment of functional performance via telehealth, including, for example, step test or timed up and go, as well as cardiorespiratory function [24]. In addition, inter-rater reliability testing remains necessary for this type of assessment in patients with long COVID.

## 5. Conclusions

The 30s-STS test proved to be a feasible test to implement in the telehealth setting and has shown a relation to fatigue, dyspnea, quality of life and pain/discomfort in a sample of community patients with long COVID. Clinicians may use this test to prescribe exercise in telehealth programs or when geographic or economic barriers prevent assessment of the physical sequelae of COVID-19.

## Figures and Tables

**Figure 1 diagnostics-13-00024-f001:**
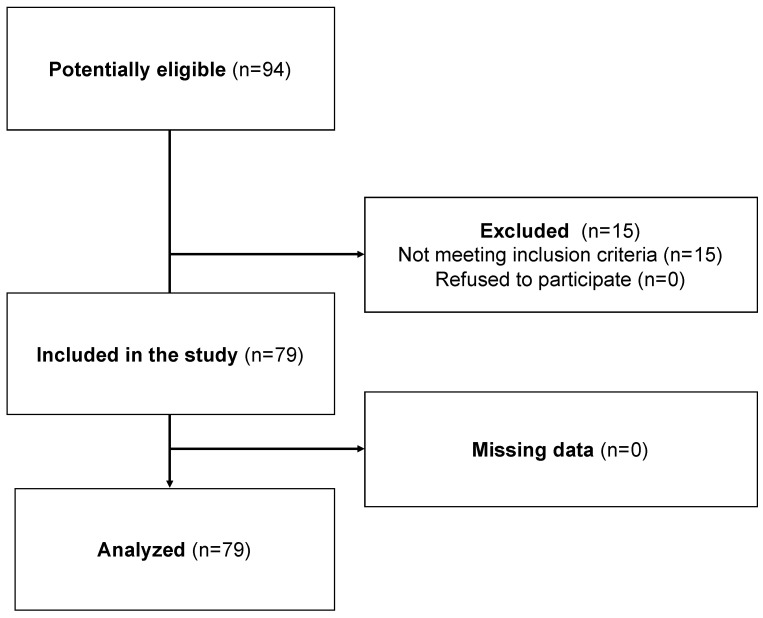
Flow-chart.

**Figure 2 diagnostics-13-00024-f002:**
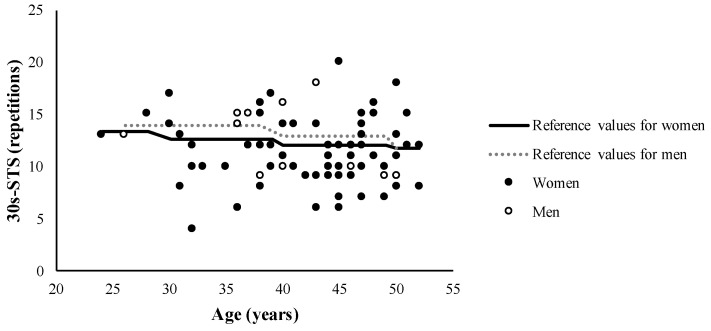
Functional performance of women and men long COVID patients vs. sex- and age-specific reference values. The reference values correspond to the sex- and age-specific zenith curves reported by Warden et al. [34] (i.e., the lower limit of the standard deviation of the mean number of repetitions in 30 s-sit-to-stand).

**Table 1 diagnostics-13-00024-t001:** Baseline characterization of patients (*n* = 79).

Characteristics	Values
Age (years)	44 (24–52)
Sex *n* (%)	
Men	11 (13.9)
Women	68 (86.1)
Post infection time (months)	17 (2–28)
Symptoms *n* (%)	
Fatigue	74 (93.7)
Dyspnea	23 (29.1)
Cognition problems	23 (29.1)
Myalgia	23 (29.1)
Headache	5 (6.3)
Cough	4 (5.1)
Smoking history *n* (%)	77 (97.5)
PCFS *n* (%)	
Grade 1	1 (1.3)
Grade 2	6 (7.6)
Grade 3	40 (50.6)
Grade 4	32 (40.5)
Grade 5	0 (0)
EQ-5D-5L	
Index score (0–1)	0.60 ± 0.23
Visual analogue scale	47.3 ± 17.1
Pain/discomfort *n* (%)	
I have no pain or discomfort	7 (8.9)
I have slight pain or discomfort	15 (19.0)
I have moderate pain or discomfort	37 (46.8)
I have severe pain or discomfort	19 (24.1)
I have extreme pain or discomfort	1 (1.3)
LCADL (10–75)	25.5 (15–68)
LCADL (%)	33.3 (20–90)
FAS (10–50)	34.7 ± 8.4
30s-STS (repetitions)	11.5 ± 3.2

Abbreviation: 30s-STS, 30 s sit-to-stand; EQ-5D-5L, European quality of life–5 dimensions–5 levels; FAS, fatigue assessment scale; LCADL, London chest activity of daily living; PCDS, Post-COVID-19 functional status. Values are mean (standard deviation), median (min-max) or *n* (%).

**Table 2 diagnostics-13-00024-t002:** Comparison between low and normal physical performance.

Characteristics	Low 30s-STS (*n* = 48)	Normal 30s-STS (*n* = 31)	*p*-Value
Age (years)	44 (9)	43 (10)	0.879
Post infection time (months)	17 (12)	18 (19)	0.948
PCFS n (%)			0.011 *
Grade 1	0 (0)	1 (3.2)	
Grade 2	1 (2.1)	5 (16.1)	
Grade 3	22 (45.8)	18 (58.1)	
Grade 4	25 (52.1)	7 (22.6)	
Grade 5	0 (0)	0 (0)	
EQ-5D-5L			
Index score (0–1)	0.51 ± 0.2	0.74 ± 0.2	<0.001 ***
Visual analogue scale	41.9 ± 14.8	55.8 ± 17.1	<0.001 ***
Pain/discomfort n (%)			0.011 *
No pain or discomfort	1 (2.1)	6 (19.4)	
Slight pain or discomfort	6 (12.5)	9 (29.0)	
Moderate pain or discomfort	25 (52.1)	12 (38.7)	
Severe pain or discomfort	15 (31.3)	4 (12.9)	
Extreme pain or discomfort	1 (2.1)	0 (0)	
LCADL (0–75)	33 (16.8)	20 (10)	<0.001 ***
LCADL (%)	44 (22.3)	26.7 (13.3)	<0.001 ***
FAS (10–50)	38 (13)	31 (12)	0.001 **
Dyspnea post 30s-STS (0–10)	4 (4)	2 (4)	0.029 *
Fatigue post 30s-STS (0–10)	7 (3)	3 (5)	<0.001 ***

Values are median (IQR) or mean ± SD. Abbreviation: 30s-STS, 30 s sit-to-stand; EQ-5D-5l, European quality of life–5 dimensions–5 levels; FAS, fatigue assessment scale; LCADL, London chest activity of daily living; PCDS, Post-COVID-19 functional status. * Statistically significant difference (*p* < 0.05); ** Statistically significant difference (*p* < 0.01); *** Statistically significant difference (*p* < 0.001).

## Data Availability

Not applicable.

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
