# Peer review of "Feasibility of the 30 s Sit-to-Stand Test in the Telehealth Setting and Its Relationship to Persistent Symptoms in Non-Hospitalized Patients with Long COVID"

_diagnostics, 2022, doi:10.3390/diagnostics13010024_

Round 1

Reviewer 1 Report

Thank you for the opportunity to review this paper. The aim of this study was to determine the feasibility of the 30s-STS test in the telehealth setting and its relationship to persistent symptoms in a sample of community patients with long COVID. The study is very timely and focuses on a really important issue. Most of the conclusions are sound. There is, however, minor issues that must be resolved before the study can be accepted for publication. Currently, it is not clear from the abstract how did you arrive to the conclusions you listed. The introduction is easy to read, however did not extend existing knowledge on this topic. See for example: Di Stefano et al. (2021). Social distancing in chronic migraine during the COVID-19 outbreak: results from a multicenter observational study. Nutrients, 13(4), 1361. In discussion,  Authors did not discuss what is novel about this research or what it offers in terms of health implications.

Author Response

In relation to the manuscript with Feasibility of the 30-s sit-to-stand test in the telehealth setting and its relationship to persistent symptoms in non-hospitalized patients with long COVID we would like to thank the reviewer for the comments on our manuscript. For sure the aspects that have been changed due to the comments will help improve the understanding of our research and its impact. Each comment has been addressed individually in this separate letter and changes in the manuscript have been highlighted so that they can be easily identified and reviewed.

Reviewer 2 Report

The manuscript reports original research on a functional test used in COVID patients in telehealth settings. In general, the paper is well-reported and conducted although I have some specific comments that require revision. Given the growing popularity and importance of alternatives to traditional/centre based assessment and rehabilitation, this paper offers a new approach in assessment.

Introduction

-Please make a better link between functional capacity importance (in COVID-19) and 30s-STS.

-Why did you use this test and not another (i.e. 6MWT )?

-Due to the outbreak of COVID-19, many rehabilitation programmes were adapted from face-to-face to distance models- Please add REF(s).

Indicate in the introduction what are the benefits of telehealth  programs and assessment compared to centre-based (removing barriers, better access for participants from rural areas, use of telehealth, suitable alternative during pandemics). These are important points that need to be detailed for the reader. (A,B,C)

A short discussion of recent meta-analyses on this topic must also be added. (D)

-Is this approach safe? Consider including a short discussion also on safety. A latest systematic review by Stefanakis et al. should be considered (E)

Methods

- Provide details about sample size calculation.

- What about the reliability and validity assessment of 30s-STS in telehealth settings.

- How did you ensure safety of exercise testing at home?

- What kind of videoconferencing setting did you use? Give us more details? How did you ensure personal data protection?

Results

***

What does this mean on Table 2?

Discussion

-‘The 30s-STS test can also be used to prescribe exercise in telehealth programmes or  when geographic or economic barriers prevent assessment of the physical sequelae of 257 COVID-19’   A strong statement was missing from the discussion. Complete an adequate discussion of covid19 and HBCR. Describe the benefits, opportunities, weaknesses, and threats. (F)

Limitations

-What kind of telemonitoring and evaluation equipment where used? Where all the same?

-Please expand future studies

Suggested References

A- Winnige P, et al. Validity and Reliability of the Cardiac Rehabilitation Barriers Scale in the Czech Republic (CRBS-CZE): Determination of Key Barriers in East-Central Europe. Int J Environ Res Public Health. 2021;18(24):13113. Published 2021 Dec 12. doi:10.3390/ijerph182413113

B- Stefanakis M, et al. Exercise-based cardiac rehabilitation programs in the era of COVID-19: a critical review. Rev Cardiovasc Med. 2021;22(4):1143-1155. doi:10.31083/j.rcm2204123

C - Nso N, et al. Comparative Assessment of the Long-Term Efficacy of Home-Based Versus Center-Based Cardiac Rehabilitation. Cureus. 2022;14(3):e23485. Published 2022 Mar 25. doi:10.7759/cureus.23485

D - Imran HM, et al. Home-Based Cardiac Rehabilitation Alone and Hybrid With Center-Based Cardiac Rehabilitation in Heart Failure: A Systematic Review and Meta-Analysis. J Am Heart Assoc. 2019;8(16):e012779. doi:10.1161/JAHA.119.012779

E- Stefanakis M, et al. Safety of home-based cardiac rehabilitation: A systematic review. Heart Lung. 2022;55:117-126. doi:10.1016/j.hrtlng.2022.04.016

F – Epstein E, et al. Cardiac Rehab in the COVID Era and Beyond: mHealth and Other Novel Opportunities. Curr Cardiol Rep. 2021;23(5):42. Published 2021 Mar 11. doi:10.1007/s11886-021-01482-7

Author Response

In relation to the manuscript with Feasibility of the 30-s sit-to-stand test in the telehealth setting and its relationship to persistent symptoms in non-hospitalized patients with long COVID we would like to thank the reviewer for the comments on our manuscript. For sure the aspects that have been changed due to the comments will help improve the understanding of our research and its impact. Each comment has been addressed individually in this separate letter and changes in the manuscript have been highlighted so that they can be easily identified and reviewed.

We are grateful for the reviewer's comments, which have helped to significantly improve the quality of the manuscript. We hope that the revised version of our manuscript has met the reviewer's expectations.

Reviewer 3 Report

Dear Authors,

The manuscript at this stage requires considerable improvements.

The study topic is interesting, however the format of the manuscript is not in accordance with the journal guidelines (text formatting, references, tables, references…), so it is suggested the correct formatting for V2.

The methodology could be more described and detailed, and the discussion improved with more text, namely with new bibliographical references.

Please review and consider the journal template and instructions for authors.

It is also important to carefully review the manuscript regarding English improvement.

Author Response

We are grateful for the reviewer's comments, which have contributed to significantly improve the quality of the manuscript. We hope that the revised version of our manuscript has met the reviewer's expectations. First, we have revised the journal guidelines to maintain proper formatting. We have also carefully revised the manuscript to improve the English. Moreover, we have added more detailed information on the methodology, regarding the method of data collection, variables, and sample size calculation (manuscript line). In addition, the introduction and discussion have been improved with new bibliographic references to justify the use of this assessment modality and the clinical relevance of the findings (manuscript line).

Round 2

Reviewer 2 Report

Most of the comments have been addressed. I endorse publication.

Author Response

We thank the reviewer for his valuable comment.

Reviewer 3 Report

Dear Authors,

Thank you for considering my suggestions and incorporating them into the manuscript. 

The manuscript improved, although the manuscript still requires considerable progress.

Below suggestions related to this last version (v2), with line indication:

2-4 – Please consider title in upper case. Please consider the journal template and instructions for authors.

5-6 – Please consider the journal template regarding line spacing.

7-16 – Please consider the journal template in text and providing the authors emails.

18 – Please describe “STS”. All abbreviations should appear in full in the first appearance in the text (please carefully review all manuscript).

17-37 – Please consider removing the subtitles.

38 – Please consider deleting.

39 – Please consider “;” instead of “,” (journal template).

123 – Please insert country in manufacturer.

121-152 – Please consider diving the text in paragraphs.

121-152 – Please describe all the details. For example, data collection conditions, procedures, who collected the data (academic background, experience), and other informations.

173-177 – Please review line spacing.

198 – Please improve figure quality.

205 – Please consider space.

206 – Please format the table considering journal template and instructions for authors.

207-210 / 2015-218 – Please consider the journal template and instructions for authors.

219-220 /231-232 - Please review line spacing.

233-240 - Please format the table, title and footer considering journal template and instructions for authors.

263 and 269 – Wrong citation format, please correct.

341 – Please remove space.

352, 352 and figures – Suggestion of inclusion in the results, not before references.

Please carefully review all the references considering the journal template. For example, ref 21 does not present page number before doi, and many other details require correction.

Please carefully revise the English before concluding V3.

Author Response

Below suggestions related to this last version (v2), with line indication:

2-4 – Please consider title in upper case. Please consider the journal template and instructions for authors.

R: We thank the reviewer for his comments. We have made the change indicated by the reviewer taking into account the journal's template and the instructions for authors.

5-6 – Please consider the journal template regarding line spacing.

R: Thank you, we made the change suggested by the reviewer.

7-16 – Please consider the journal template in text and providing the authors emails.

R: We thank the reviewer for his comments. We have made the change indicated by the reviewer taking into account the journal's template and the instructions for authors.

18 – Please describe “STS”. All abbreviations should appear in full in the first appearance in the text (please carefully review all manuscript).

R: Thank you for your important comment, we have added this description as you indicated.

17-37 – Please consider removing the subtitles.

R: Thank you, we made the change suggested by the reviewer.

38 – Please consider deleting.

R: Thank you, we made the change suggested by the reviewer.

39 – Please consider “;” instead of “,” (journal template).

R: Thank you, we made the change suggested by the reviewer.

123 – Please insert country in manufacturer.

R: Thank you, we made the change suggested by the reviewer.

121-152 – Please consider diving the text in paragraphs.

R: Thank you, we made the change suggested by the reviewer.

121-152 – Please describe all the details. For example, data collection conditions, procedures, who collected the data (academic background, experience), and other informations.

R: We thank the reviewer for his valuable comment. We have added the following information:  “Data were collected by four PhD physiotherapists (C.F-R., F.M.M-A., A.A.-G. and D.H-G.) with more than five years of clinical experience. All assessors received prior training to standardise the assessment.”

173-177 – Please review line spacing.

R: Thank you, we have corrected this.

198 – Please improve figure quality.

R: We thank the reviewer for his comment. The quality of the figures has been improved.

205 – Please consider space.

R: Thank you, we have corrected this.

206 – Please format the table considering journal template and instructions for authors.

R: We thank the reviewer for his comments. We have made the change indicated by the reviewer taking into account the journal's template and the instructions for authors

207-210 / 2015-218 – Please consider the journal template and instructions for authors.

R: We have made the change indicated by the reviewer taking into account the journal's template and the instructions for authors

219-220 /231-232 - Please review line spacing.

R: Thank you, we have corrected this.

233-240 - Please format the table, title and footer considering journal template and instructions for authors.
R: We thank the reviewer for his comments. We have made the change indicated by the reviewer taking into account the journal's template and the instructions for authors

263 and 269 – Wrong citation format, please correct.

R: Thank you, we have corrected this.

341 – Please remove space.

R: Thank you, we have corrected this.

352, 352 and figures – Suggestion of inclusion in the results, not before references.

R: We thank the reviewer for his comments. These results were presented as supplementary material (Appendix A) and were subsequently added here by the editorial staff that generated the evidence. Therefore, we have now removed it from the manuscript.

Please carefully review all the references considering the journal template. For example, ref 21 does not present page number before doi, and many other details require correction.

Please carefully revise the English before concluding V3.

R: We thank the reviewer for his valuable comments. We have carefully checked all references taking into account the journal template.

In addition, we have carefully checked the English as suggested.

Round 3

Reviewer 3 Report

Dear Authors,

Thank you for considering my suggestions and incorporating them into the manuscript. Congratulations for the work performed within the scope of this manuscript.

Below small suggestions related to this last version (v3), with line indication.

8-18 - The number of affiliations should be changed considering the journal template and instructions for authors.

21,39 - The removal of subtitles in abstract is suggested.

43 - Please consider deleting the space.

207 - Please consider deleting the space.

217-219 / 242-246 – Please consider journal template and instructions for authors in the table footnotes (legend) – below the table.

241 - Please consider deleting the space.

247 - Please consider deleting the space.

332 – Please consider the standardization of text format “step test or Timed Up and Go”.

It is suggested a final double check of the English throughout the manuscript and in the references format.

Author Response

We thank the reviewer for his comments. We have made the change indicated by the reviewer taking into account the journal's template and the instructions for authors. We hope that the revised version of our manuscript has met the reviewer's expectations.
